# Design and Development of an Active Suspension System Using Pneumatic-Muscle Actuator and Intelligent Control

**I-Hsum Li [1]** and **Lian-Wang Lee [2,*]**

[1] Department of Mechanical and Electro-Mechanical Engineering, Tamkang University, No. 151, Yingzhuan Road, Tamsui District, New Taipei City 25137, Taiwan; ihsumlee@gmail.com

[2] Department of Mechanical Engineering, National Chung Hsing University, No. 145, Xingda Road, South District, Taichung City 40227, Taiwan

[*] Correspondence: leelw@dragon.nchu.edu.tw; Tel.: +886-4-2284-0433 (ext. 420)

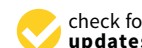

**Featured Application: Vehicle Suspension.**

**Abstract:** A pneumatic muscle is a cheap, clean, and high-power active actuator. However, it is difficult to control due to its inherent nonlinearity and time-varying characteristics. This paper presents a pneumatic muscle active suspension system (PM-ASS) for vehicles and uses an experimental study to analyze its stability and accuracy in terms of reducing vibration. In the PM-ASS, the pneumatic muscle actuator is designed in parallel with two MacPherson struts to provide a vertical force between the chassis and the wheel. This geometric arrangement allows the PM-ASS to produce the maximum force to counter road vibration and make the MacPherson struts generate significant improvement. In terms of the controller design, this paper uses an adaptive Fourier neural network sliding-mode controller with $H_\infty$ tracking performance for the PM-ASS, which confronts nonlinearities and time-varying characteristics. A state-predictor is used to predict the output error and to provide the predictions for the controller. Experiments with a rough concave-convex road and a two-bump excitation road use a quarter-car test rig to verify the practical feasibility of the PM-ASS, and the results show that the PM-ASS gives an improvement the ride comfort.

**Keywords:** active suspension systems; pneumatic muscles; sliding mode control; fourier neural network; $H_\infty$ tracking performance

## 1. Introduction

Active suspension systems (ASS) [1–4] have been the subject of much development in the car industry over decades. To provide comfort ride and road holding for driving, ASS actively controls the vertical movement of the wheels when the road terrain changes. Therefore, ASS actively reduces road vibrations and gives a smoother ride experience for passengers than passive/conventional suspension systems because the design of a passive suspension system compromises between three conflicting criteria: road handling, road carrying, and passenger comfort [5,6].

To improve road handling and passenger comfort, ASS uses actuators to provide additional force between the sprung mass and the wheels to control the attitude of the vehicle. ASS also reduces vertical vibration in cameras that are mounted on a vehicle, thus, stable camera sequences are ensured for image processing [1,2]. Therefore, ASS is eminently suited to the needs of self-driving ground vehicles. The most-commonly used actuators for an ASS are pneumatic, linear electromagnetic, or hydraulic actuators. For a classical and conventional configuration, these actuators are commonly designed to be used in conjunction spring dampers to create an active suspension, which gives an additional vertical

force between the chassis and the wheels [7]. A hydraulic actuator is one of the most commonly-used linear actuators for ASS, because it is tough and has a high power-to-weight ratio. It also maintains a constant force and torque without the need for extra fluid or pressure. Huang et al. [7] designed a hydraulic-driven ASS. The hydraulic ASS uses a model-free adaptive sliding controller to suppress vehicle vibration due to a 20 mm-high bump on a test-rig. Lin et al. [8] presented a hydraulic-driven ASS that is regulated using an enhanced fuzzy sliding controller, which effectively suppresses vibration. However, a hydraulic ASS creates environmental pollutions because of the possibility of fluid leaks.

Recently, a serial active variable geometry suspension (SAVGS) [9–11] has been theoretically and practically demonstrated on its feasibility for active suspensions. SAVGS uses a device between the spring-damper and the adjacent body. The device is mounted in series with the spring-damper and contains an electro-mechanical actuator that controls the orientation and elongation of the strut. Pneumatic actuators are also eminently suited to use in an ASS. A pneumatic active suspension can be modeled as a system with a spring, a passive damper, and a pneumatic active component. A pneumatic suspension has advantages over a spring-damper suspension in that it gives a soft ride at low speed while driving on flat terrain but allows good control while driving on rough terrain and it provides a force that allows the height of the vehicle's chassis to be adjusted to suit particular terrain.

Winfred et al. [12] designed a prototype of a pneumatic ASS for teaching students. The pneumatic ASS consists of a wheel, coil springs, a pneumatic actuator for active damping, and an alternating current (ac) motor to simulate various road conditions. Nieto et al. [13] presented an analytical model of a pneumatic suspension and tuned its behavior to operate an ASS. An air spring can also be used as a pneumatic actuator for an ASS. It uses an airbag as a spring to adjust the height of a vehicle's chassis. An air spring is both a spring and a pneumatic actuator, thus, it gives good control for load carrying, stance, tenability, and handling. Alireza [14] developed a quarter-car suspension system that uses an air spring active actuator to give improved vibration isolation in the low-frequency range. Graf et al. [15] designed a pneumatic push-pull actuator to isolate vibration in a vehicle. The push-and pull-functionality is separated into three actuators: an active air spring for the push direction and two actuators that are fluidic muscles for the pull direction.

The pneumatic muscle (PM) [16,17] was developed in the 1950s and is suited to devices that are worn. The retraction strength of a PM depends on the strength of the individual fibers in the woven shell. A PM has several advantages in that it is highly compliant, low-cost, lightweight, and safe. It is used in rehabilitation engineering, which requires great compliance and a high level of safety for patients [18]. A study by Ostasevicius et al. [19] demonstrates a pneumatic-muscle ASS that uses one FESTO artificial muscle (MAS-20-N400) parallel with two springs to provide a vertical force for vehicles. The study shows that the PM-driven ASS can vibrate at a 40 mm-amplitude at the frequency of 1.5 Hz and vibrates at 10–25 mm-amplitude within the frequency range of 4–8 Hz. The length of the PM is 400 mm and air flows into the PM from one side only. The report demonstrates that the main challenge in designing a PM-driven ASS is the inherent non-linear deformation characteristics.

A PM-ASS has never used in commercial vehicles; thus, this study develops a test-rig for a PM-driven ASS and determines its ability to reduce vibration. The test-rig is designed as a quarter-car and comprises a FESTO PM, which is installed between the sprung mass and the wheel and two MacPherson struts provide additional vertical force. The test-rig simulates different driving conditions in terms of the speed of rotation of the wheel and road terrain. For ASS control, an adaptive Fourier-series-based adaptive sliding-mode controller (FSB-ASMC) with $H_\infty$ tracking performance is used to compensate for all possible nonlinearities, uncertainties, and external vibration. A Fourier neural network (Fourier-NN) [20,21] combines Fourier analysis and neural network theory. A Fourier-NN uses a family of complex Fourier functions as an activation function $\sigma_i(\cdot)$, which decomposes a signal into its component frequencies with different amplitudes and phases. The FSB-ASMC also uses $H_\infty$ tracking for the adaptive sliding-mode control scheme, which protects the systems against approximation errors, disturbances and unmodeled dynamics and guarantees the desired $H_\infty$ tracking performance

for the overall system. It also significantly reduces the control chatter that is inherent in a conventional sliding mode control system.

The main contributions of this paper are as follows:

1. This paper uses a pneumatic muscle actuator with two MacPherson struts to increase the isolation of vibration and show its feasibility on a quarter-car test rig.
2. A fully-functional test-rig for a quarter car is developed for examining the PM-ASS which has a road profile generator to provide real road conditions.
3. Using the orthogonal activation basis functions of Fourier series, Fourier-NN has a clear physical meaning and readily determined structure, which gives it an advantage over conventional NNs. Because of the orthogonality of the basis functions, the Fourier-NN converges very fast, which means it is suited to real-time implementation.
4. A state-predictor $GM(1,1)$ is used to predict the output error and to provide the predictions for the controller.
5. Four different control strategies are conducted to analyze the stability and accuracy in terms of reducing vibration, which are MacPherson Strut, FSASMC, FSB-ASMC+ $H_\infty$, and GP-FSB-ASMC+ $H_\infty$.

## 2. Description of the Mathematical Model

### 2.1. Mathematical Model of the PM

Figure 1 shows the design of the PM. It is made of natural rubber with fibers wrapped inside and (see Figure 1a) metal fittings attached at each end. The PM changes shape when the pressure of the air that flows into the interior of the rubber tube is changed. Injecting compressed air into the rubber tube expands the PM and produces a force that drags the sprung mass downward (see Figure 1b). Otherwise, the sprung mass returns to its original shape if the PM releases air. For simplicity, the reversible physical deformation during contraction and expansion produces a linear motion. According to the standard static physical model [22,23], (see Figure 2), the length and the diameter of the PM are formulated as:

$$L = l \cos \theta \tag{1}$$

and:

$$D = \frac{l \sin \theta}{n\pi} \tag{2}$$

where $L$ represents the actual length of PM, $L_0$ represents the original length of PM, $l$ represents the length of the thread, $n$ represents the number of turns in the thread, $D$ represents the diameter, and $\theta$ represents the angle that the threads make with a horizontal axis.

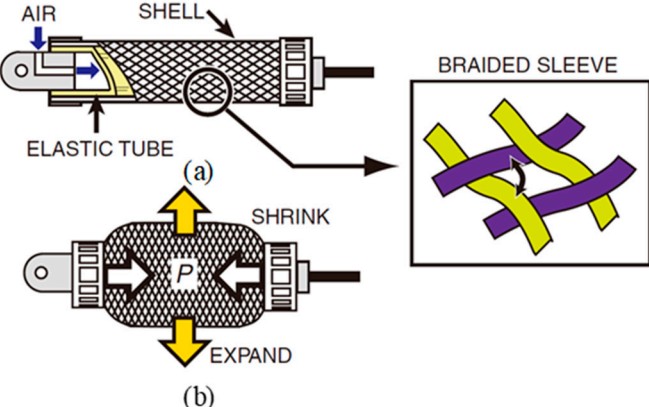

**Figure 1.** Design of the pneumatic muscle (PM) (**a**) Components of the PM; (**b**) Actions of the PM.

Considering the PM as a cylindrical shape, its volume is defined as:

$$V = \frac{\pi D^2 L}{4} \tag{3}$$

Substituting Equations (1) and (2) into (3) yields:

$$V = \frac{l^3 \left(1 - \cos^2 \theta\right) \cos \theta}{4\pi n^2} \tag{4}$$

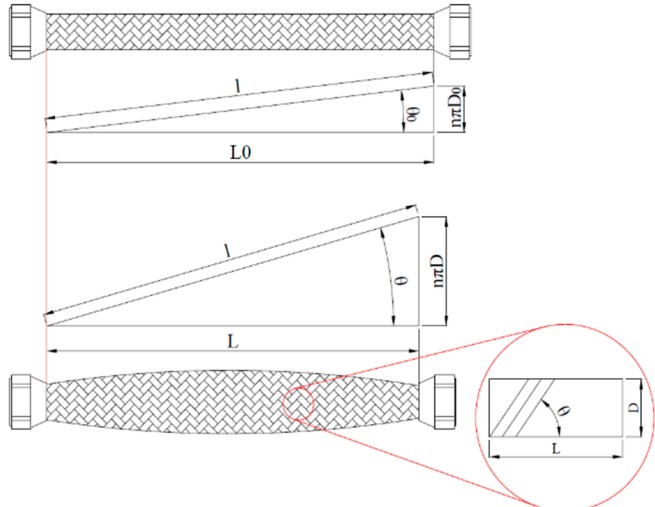

**Figure 2.** Model of the PM.

Because the active area of the cylinder changes over time, the volume of the PM changes with time. The active area is expressed as $\overline{A} = dV/dL$, which is:

$$\overline{A} = \frac{3L^2 - l^2}{4\pi n^2} \tag{5}$$

According to the Principle of the Conservation of Energy [24], the geometric force [22] is the product of the pressure and the change in the volume with respect to the length, which is:

$$F_a = -(P - P_e)\frac{dV}{dL} \tag{6}$$

where $P$ represents the pressure inside the PM and $P_e$ represents atmospheric pressure. Substituting Equation (5) into (6) yields:

$$F_a = -(P - P_e)\frac{3L^2 - l^2}{4\pi n^2} \tag{7}$$

A proportional pressure regulator (PPR) is used to regulate the pressure $P$ inside the PM. This is modeled by a first order transfer function, which is:

$$P = \frac{K_v}{T_s s + 1}u \tag{8}$$

where $K_v$ and $T_s$ respectively represent the constant gain and the time constant for the PM, and $u$ represents the input voltage. Substituting Equation (8) into (7) yields:

$$F_a = -(\frac{K_v}{T_s s + 1}u - P_e)\frac{3L^2 - l^2}{4\pi n^2} \tag{9}$$

**Remark 1.** *In general, a PPR exhibits complex dynamic behavior and must be modeled by a high order nonlinear differential equation. Fortunately, for the PM-ASS, the bandwidth for a PPR is much wider than that for a PM; thus, the dynamics of a PPR can be modeled as a first order linear differential equation, as shown in Equation (8).*

### 2.2. Mathematical Model of a Quarter Car

Figure 3 shows an illustration of a quarter-car with a PM-ASS, in which the PM-ASS has two MacPherson struts and a PM, as shown in Figure 3a. For simplicity, the mathematical models of the two MacPherson struts are combined in one model, thus, they are expressed as one damper, one spring, and an internal frictional force. The dynamics of a tire are modeled as a spring if the tire fully contacts with the road surface. In this condition, the quarter-car is modeled as a system with 2-DOF (two-degrees-of-freedom). The frictional force of the piston in the shock absorber prevents the smooth travel of the MacPherson struts so the dynamic behavior of the PM-ASS must consider the frictional force. The dynamic equation for a quarter-car with a PM-ASS is:

$$M_s\ddot{Z}_s = K_s(Z_u - Z_s) + B_s(\dot{Z}_u - \dot{Z}_s) + F_a - F_\mu \text{sgn}(\dot{Z}_u - \dot{Z}_s) \tag{10}$$

and:

$$M_u\ddot{Z}_u = -K_s(Z_u - Z_s) - B_s(\dot{Z}_u - \dot{Z}_s) + K_t(Z_r - Z_u) - F_a + F_\mu \text{sgn}(\dot{Z}_u - \dot{Z}_s) \tag{11}$$

where $F_\mu$ is the frictional force, $F_a$ is the force that is exerted by the PM, $Z_r$ is the variation in the position of the road surface, $Z_s$ is the displacement of the sprung mass, $Z_u$ is the displacement of the un-sprung mass, $M_s$ is the weight of the rigid body (sprung mass), $M_u$ is the weight of the un-sprung mass, $K_t$ is a spring constant, and $B_s$ is a damping constant. If $x_1 = Z_s$, $x_2 = \dot{Z}_s$, $x_3 = Z_u$, $x_4 = \dot{Z}_u$, and $x_5 = P$, then manipulating Equations (8)–(11) with $x_1$ to $x_5$ yields a dynamic equation:

$$
\begin{aligned}
\dot{x}_1 &= x_2 \\
\dot{x}_2 &= -\frac{1}{M_s}[K_s(x_1 - x_3) + B_s(x_2 - x_4) - x_5\frac{3L^2-l^2}{4\pi n^2} + \frac{3L^2-l^2}{4\pi n^2}P_e + F_\mu \text{sgn}(x_2 - x_4)] \\
\dot{x}_3 &= x_4 \\
\dot{x}_4 &= \frac{1}{M_u}[K_s(x_1 - x_3) + B_s(x_2 - x_4) - K_t(x_3 - Z_r) - x_5\frac{3L^2-l^2}{4\pi n^2} + \frac{3L^2-l^2}{4\pi n^2}P_e + F_\mu \text{sgn}(x_2 - x_4)] \\
\dot{x}_5 &= \frac{1}{T_s}(-x_5 + K_V u) \\
y &= x_1
\end{aligned}
\tag{12}
$$

Equation (12) can further be expressed as its matrix form, as given by:

$$
\begin{aligned}
\dot{\mathbf{x}} &= f(\mathbf{x}) + g(\mathbf{x})u \\
y &= h(\mathbf{x}) = x_1
\end{aligned}
\tag{13}
$$

where:

$$
f(\mathbf{x}) =
\begin{bmatrix}
x_2 \\
-\frac{1}{M_s}[K_s(x_1 - x_3) + B_s(x_2 - x_4) - x_5\frac{3L^2-l^2}{4\pi n^2} + \frac{3L^2-l^2}{4\pi n^2}P_e] + F_\mu sgn(x_2 - x_4) \\
x_4 \\
\frac{1}{M_u}[K_s(x_1 - x_3) + B_s(x_2 - x_4) - K_t(x_3 - Z_r) - x_5\frac{3L^2-l^2}{4\pi n^2} + \frac{3L^2-l^2}{4\pi n^2}P_e] + F_\mu sgn(x_2 - x_4) \\
\frac{-x_5}{T_s}
\end{bmatrix}
\tag{14}
$$

and:

$$g(\mathbf{x}) = \begin{bmatrix} 0 \\ 0 \\ 0 \\ 0 \\ \frac{K_V}{T_s} \end{bmatrix} \tag{15}$$

Since $L_{g(\mathbf{x})}L^{\kappa}_{f(\mathbf{x})}h(\mathbf{x}) = 0$ for all $\kappa < 3$, and $L_{g(\mathbf{x})}L^{\kappa}_{f(\mathbf{x})}h(\mathbf{x}) \neq 0$ for all $\kappa \geq 3$ $(\forall x)$, the relative degree of the system is 3. Using input-output Feedback linearization, Equation (13) is transformed into a canonical form after simple manipulation, which is:

$$y^{(3)} = L^3_{f(\mathbf{x})}h(\mathbf{x}) + L_{g(\mathbf{x})}L^2_{f(\mathbf{x})}h(\mathbf{x})u = F(\mathbf{x}) + G(\mathbf{x})u, \tag{16}$$

where:

$$F(\mathbf{x}) = -\frac{1}{M_s}\{K_s(x_2 - x_4) - B_s(\frac{1}{M_s} + \frac{1}{M_u})\{K_s(x_1 - x_3) + B_s(x_2 - x_4) - \frac{\{3[L_0 + (x_1 - x_3)]^2 - l^2\}}{4\pi n^2}x_5 + \\ \frac{\{3[L_0 + (x_1 - x_3)]^2 - l^2\}}{4\pi n^2}P_e + F_\mu \mathrm{sgn}(x_2 - x_4)]\} + \frac{B_s K_t}{M_u}x - \frac{B_s K_t}{M_u}Z_r + \frac{\{3[L_0 + (x_1 - x_3)]^2 - l^2\}}{4\pi n^2 T_s}x_5 - \\ \frac{\{6[L_0 + (x_1 - x_3)](x_2 - x_4)\}}{4\pi n^2}x_5 + \frac{\{6[L_0 + (x_1 - x_3)]^2(x_2 - x_4)\}}{4\pi n^2}P_e + \dot{F}_\mu \mathrm{sgn}(x_2 - x_4)\} \tag{17}$$

and:

$$G(\mathbf{x}) = \frac{K_v}{M_s T_s} \frac{\left\{3[L_0 + (x_1 - x_3)]^2 - l^2\right\}}{4\pi n^2} \tag{18}$$

The states are selected as $\mathbf{z}(t) = \begin{bmatrix} z_1(t) & z_2(t) & z_3(t) \end{bmatrix}^T = \begin{bmatrix} y(t) & \dot{y}(t) & \ddot{y}(t) \end{bmatrix}^T \in R^3$, thus, Equation (16) can be expressed as:

$$\begin{aligned} \dot{z}_1 &= z_2 \\ \dot{z}_2 &= z_3 \\ \dot{z}_3 &= \overline{F}(\mathbf{z}) + \overline{G}(\mathbf{z})u \\ y &= z_1 \end{aligned} \tag{19}$$

where the output $z_1$ is the displacement of the sprung mass, i.e., $z_1 = x_1 = y$, $\overline{F}(\mathbf{z})$ and $\overline{G}(\mathbf{z})$ cannot be assumed to be given continuous functions because they might contain unknown parameters, nonlinearities and uncertainties. It is noted that the functions $\overline{F}(\mathbf{z})$ and $\overline{G}(\mathbf{z})$ are unknown so it is impossible to determine a stable ideal control law for a quarter-car with a PM-ASS, as given by:

$$u = 1/\overline{G}(\mathbf{z})[\sum_{i=1}^{2} k_i e^{(i)} - \overline{F}(\mathbf{z}) + z_d^{(3)}] \tag{20}$$

Therefore, a Fourier neural network (FNN) is adapted to approximate the unknown functions $\overline{F}(\mathbf{z})$ and $\overline{G}(\mathbf{z})$; thus, a stable and robust control law can be derived for a quarter-car with a PM-ASS.

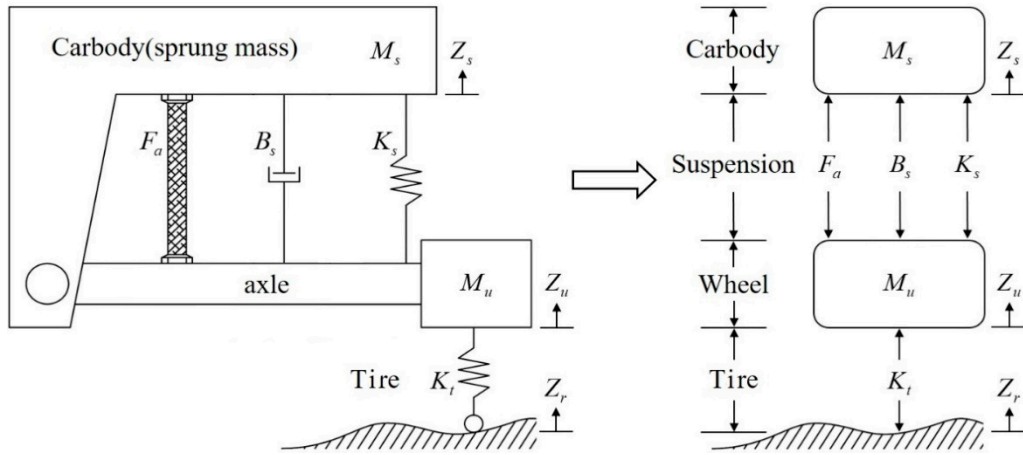

**Figure 3.** Illustration of a quarter-car with a PM- active suspension system (ASS).

## 3. Adaptive Fourier Neural Network Sliding-Mode Controller (FSB-ASMC) with $H_\infty$ Tracking Performance

The PM-ASS features internal parametric uncertainties and external disturbances $d(t)$ that arise from different unknown road terrains while driving; thus, the FSB-ASMC with $H_\infty$ tracking performance was used for this study. While considering external disturbances $d(t)$, the mathematical model for the Quarter Car with PM-ASS (see Equation (19)) is changed to:

$$\dot{z}_3 = \overline{F}(\mathbf{z}) + \overline{G}(\mathbf{z})u + d(t) \tag{21}$$

where $\mathbf{z}(t) = [\ y(t)\quad \dot{y}(t)\quad \ddot{y}(t)\ ]^T \in R^3$ is a vector of state s, which are assumed to be measurable; $u(t) \in R$ and $y(t) \in R$ are the control input and system output, respectively, $d(t)$ is the bounded external disturbance, i.e., $|d(t)| \le d^u$, $\overline{F}(\mathbf{z})$ and $\overline{G}(\mathbf{z})$ are smooth unknown functions with uncertain time-varying parameter. Without loss of generality, $\overline{G}(\mathbf{z})$ can be assumed to be strictly positive; i.e., $\overline{G}(\mathbf{z}) \ge G^l > 0$. It is assumed that there exists a solution for (21) and the order of the nonlinear system (21) is known.

### 3.1. Sliding Mode Control

Sliding mode control can be used to reject uncertainties and external disturbance. Generally, it assumes that $\overline{F}(\mathbf{z})$ and $\overline{G}(\mathbf{z})$ are given. Here, we define a sliding mode control law for the quarter-car with a PM-ASS while the system contains external disturbances. The output tracking error is written as:

$$e = y - y_d \tag{22}$$

where $y_d(t)$ is a reference signal, which represents road conditions in this paper. A switch surface for an nth-order system is defined as:

$$s = c_1 e + c_2 \dot{e} + \ddot{e} \tag{23}$$

where $c_i$ are chosen such that $\sum\limits_{i=1}^{3} c_i l^{i-1}$ is a Hurwitz polynomial in which $l$ is a Laplace operator. Equation (23) shows that:

$$\ddot{e} = -c_1 e - c_2 \dot{e} + s \tag{24}$$

If $\mathbf{e} = [e, \dot{e}]^T = [e_1, e_2]^T$, the error dynamics are:

$$\dot{\mathbf{e}} = \mathbf{A}_1 \mathbf{e} + [0, s]^T \tag{25}$$

where $\mathbf{A}_1 = \begin{bmatrix} 0 & 1 \\ -c_1 & -c_2 \end{bmatrix}$.

**Lemma 1.** *For a nonlinear system (21) with the given nonlinear functions $\overline{F}(\mathbf{z})$ and $\overline{G}(\mathbf{z})$, if the control input is:*

$$u = \frac{-\overline{F}(\mathbf{z}) - \sum\limits_{i=1}^{2} c_i e_{i+1} - \sum\limits_{i=1}^{2} p_{2i} e_i + y_d^{(3)} - k\mathrm{sgn}(s)}{\overline{G}(\mathbf{z})} \tag{26}$$

*and $\mathbf{P} > 0$, $\mathbf{P} \in \mathbf{R}^{2 \times 2}$ satisfies the Lyapunov matrix equation:*

$$\mathbf{A}_1^T \mathbf{P} + \mathbf{P} \mathbf{A}_1 = -\mathbf{Q_l} \tag{27}$$

*where s is the sliding surface defined in Equation (23), $p_{2i}$ are elements of $\mathbf{P}$, $k > 0$, and $\mathbf{Q_l} > 0$ is given, then $s \rightarrow 0$ and $e \rightarrow 0$ as $t \rightarrow \infty$.*

**Proof:** See Appendix A. □

However, $\overline{F}(\mathbf{z})$ and $\overline{G}(\mathbf{z})$ for a quarter-car with a PM-ASS are generally uncertain rather than given, thus, there is no control input for Equation (26). The constant $k$ in Equation (26) is designed to attenuate the uncertainties. However, a larger value for $k$ produces increased chatter. To relax these two constraints, a Fourier Neural Network (Fourier-NN) is used to approximate the unknown functions $\overline{F}(\mathbf{z})$ and $\overline{G}(\mathbf{z})$ to eliminate model dependency. $H_\infty$ tracking design is also used to compensate for the approximation error, which gives improved control performance and reduced chatter.

*3.2. Description of Fourier Neural Networks*

In this study, a Fourier-NN is used to approximate the unknown nonlinear functions with uncertainties, $\overline{F}(\mathbf{z})$ and $\overline{G}(\mathbf{z})$, in order to provide a stable control law for the PM-ASS. A Fourier-NN is eminently suited to the modeling and decomposition of a nonlinear function. It uses a Fourier series as the basic functions and allows much better convergence for the approximation. It is noted that the number of the hidden layers of the Fourier-NN depends on the bandwidth of the system. The weigh vector for a Fourier-NN is acquired used the Lyapunov stability theorem. The output function for a Fourier-NN is:

$$p(\chi) = \mathbf{W}^T \mathbf{q}(\chi) \tag{28}$$

where $\chi$ and $p(\chi)$, respectively, represent the input and the output for the Fourier-NN, $\mathbf{q}(x) \equiv \left[\cos(w_0\chi), \quad \cos(w_1\chi), \quad \sin(w_1\chi), \ldots, \cos(w_M\chi), \quad \sin(w_M\chi)\right]^T$ is a family of orthogonal Fourier activation functions, and $\mathbf{W} \equiv \left[W_0 \quad W_1 \quad \ldots \quad W_{2M-1} \quad W_{2M}\right]^T$ is a vector for network weights.

**Remark 2.** *The approximation error for a Fourier-NN is bounded and is expressed as:*

$$\varepsilon_n(\chi) = |f(\chi) - p(\chi)| \leq \sum_{i>n}(|w_i| + |w_{i+1}|); \ n = 2M + 1 \tag{29}$$

*Clearly, the approximation error $\varepsilon_n(t)$ is eliminated as $n \rightarrow \infty$. This implies that $f(\chi)$ is approximated as long as n is sufficiently large.*

**Remark 3.** *If a nonlinear function $f(t)$ is a non-periodical function, which is written as:*

$$f(t) = \int_{-\infty}^{\infty} F(w)e^{jwt}dw, \tag{30}$$

where $F(w) = \int_{-\infty}^{\infty} f(t)e^{-jwt}dt$, then by applying Shannon's theory in the frequency domain, Equation (30) is equivalent to the partition of a non-periodical function with an appropriate window in the time domain. This is written in discrete form as:

$$f(t) = \int_{-\infty}^{\infty} F(w)e^{jwt}dw = \sum_{n=-M}^{M} X(n\Omega)e^{jn\Omega t}\Delta w, \tag{31}$$

where $\Omega = w_n/n$ is the base frequency that is acquired by discretizing $w$. From an engineering point of view, the control objective is defined as a limited time interval of $[0, T]$, thus, the base frequency is simply selected as $\Delta w = 2\pi/T$.

### 3.3. Design of the FSB-ASMC with $H_\infty$ Tracking Performance

Because the control voltage for the PM-ASS is specified and bounded, all of the states, which are real-world physical signals, are bounded. This implies that $\overline{F}(\mathbf{z}(t))$ and $\overline{G}(\mathbf{z}(t))$ are bounded and meet the Dirichlet conditions. If time is the input for the FNN, i.e., $\chi = t$ in (28), and two Fourier-NNs are used to respectively approximate $\overline{F}(\mathbf{z}(t))$ and $\overline{G}(\mathbf{z}(t))$, Equation (21) is rewritten as:

$$\begin{aligned}\dot{z}_3 &= (\mathbf{W}_{\overline{F}}^T\mathbf{q}_{\overline{F}}(t) + \varepsilon_{\overline{F}}(t)) + (\mathbf{W}_{\overline{G}}^T\mathbf{q}_{\overline{G}}(t) + \varepsilon_{\overline{G}}(t))u \\ &= \mathbf{W}_{\overline{F}}^T\mathbf{q}_{\overline{F}}(t) + \mathbf{W}_{\overline{G}}^T\mathbf{q}_{\overline{G}}(t)u + w_t\end{aligned} \tag{32}$$

where $\mathbf{q}_{\overline{F}}(t)$ and $\mathbf{q}_{\overline{G}}(t)$ are the family of orthogonal Fourier activation function vector, $\mathbf{W}_{\overline{F}}$ and $\mathbf{W}_{\overline{G}}$ are the weight vectors, and $w_t = \varepsilon_{\overline{F}}(t) + \varepsilon_{\overline{G}}(t)u$ is the lumped uncertainty subject to $|w_t| \le w_t^u$, where $w_t^u$ is a positive constant. The terms $\mathbf{W}_{\overline{F}}^T\mathbf{q}_{\overline{F}}$ and $\mathbf{W}_{\overline{G}}^T\mathbf{q}_{\overline{G}}$ produce an optimal mean-square approximation to the uncertain time-varying functions $\overline{F}(\mathbf{z}(t))$ and $\overline{G}(\mathbf{z}(t))$.

**Assumption 1.** *The lumped uncertainty in (19) is assumed such that $w \in L_2[0, T]$, $\forall T \in [0, \infty)$.*

**Assumption 2.** *The internal dynamics of the system (19) that uses adaptive sliding-mode control law are stable.*

**Theorem 1.** *If a system can be expressed as a form of Equation (21) with unknown nonlinear time-varying functions $\overline{F}(z(t))$ and $\overline{G}(z(t))$ and has finite bandwidth. The sliding-mode control input is:*

$$u = \frac{-\hat{\mathbf{W}}_{\overline{F}}^T\mathbf{q}_{\overline{F}}(t) - \sum_{i=1}^{2} c_ie_{i+1} - \sum_{i=1}^{2} p_{2i}e_i + y_d^{(3)} - \frac{s}{2\sigma^2}}{\hat{\mathbf{W}}_{\overline{G}}^T\mathbf{q}_{\overline{G}}(t)} \tag{33}$$

where $\hat{\mathbf{W}}_{\overline{F}}^T$ and $\hat{\mathbf{W}}_{\overline{G}}^T$ are the estimates of $\mathbf{W}_{\overline{F}}^T$ and $\mathbf{W}_{\overline{G}}^T$, respectively. According to **Lemma 1**, $\mathbf{P} > 0$, $\mathbf{P} \in R^{2\times2}$ is selected to satisfy the Lyapunov matrix equation:

$$\mathbf{A}_1^T\mathbf{P} + \mathbf{P}\mathbf{A}_1 = -\mathbf{Q}_1 \tag{34}$$

where $s$ is the sliding surface that is defined in Equation (23), $p_{2i}$ are elements of $\mathbf{P}$, $\sigma$ is a designed constant, $\mathbf{Q}_1 > 0$ is given and the adaptive laws are:

$$\dot{\hat{\mathbf{W}}}_{\overline{F}} = \eta_1 s\mathbf{q}_{\overline{F}}(t) \tag{35}$$

and:

$$\dot{\hat{\mathbf{W}}}_{\overline{G}} = \eta_2 s\mathbf{q}_{\overline{G}}(t)u \tag{36}$$

*where $\eta_1$ and $\eta_2$ ($\eta_1 > 0$ and $\eta_2 > 0$) are the adaptation gain matrices. Therefore, the $H_\infty$ tracking performance for the overall system must satisfy: (see (37))*

$$\frac{1}{2}\int_0^T \mathbf{e}^T(\tau)\mathbf{Q_I}\mathbf{e}(\tau)d\tau \le \frac{1}{2}s^2(0) + \frac{1}{2}\mathbf{e}^T(0)\mathbf{Ae}(0) + \frac{1}{2}\widetilde{\mathbf{W}}_{\overline{F}}^T(0)\mathbf{\Gamma}_1^{-1}\widetilde{\mathbf{W}}_{\overline{F}}(0) + \frac{1}{2}\widetilde{\mathbf{W}}_{\overline{G}}^T(0)\mathbf{\Gamma}_2^{-1}\widetilde{\mathbf{W}}_{\overline{G}}(0) + \frac{1}{2}\sigma^2\int_0^T w^2(\tau)d\tau \quad (37)$$

*where $\widetilde{W}_{\overline{F}} = W_{\overline{F}} - \hat{W}_{\overline{F}}$ and $\widetilde{W}_{\overline{G}} = W_{\overline{G}} - \hat{W}_{\overline{G}}$.*

**Proof:** See Appendix B. □

**Remark 4.** *For a specified set of initial conditions, $e(0) = 0$, $s(0) = 0$, $\hat{W}_{\overline{F}}(0) = \hat{W}_{\overline{F}}^*(0)$ and $\hat{W}_{\overline{G}}(0) = \hat{W}_{\overline{G}}^*(0)$, and if the matrix $Q_I$ is set to an identity matrix, the control performance satisfies:*

$$\frac{\|\mathbf{e}\|_2}{\|w_t\|_2} \le \sigma \tag{38}$$

*where $\|w_t\|_2^2 = \int_0^T w_t^2(\tau)d\tau$ and $\|e\|_2^2 = \int_0^T e^T(\tau)e(\tau)d\tau$. In other words, an arbitrary attenuation level is achieved if $\sigma$ is properly chosen. Equation (38) also shows that $\|e\|_2 \le \sigma\|w_t\|_2$, thus, the bearing of the system (21) on the lumped uncertainty is stable. According to **Assumption 2**, the system (13) is stable.*

### 3.4. Design Procedures

Figure 4 shows the overall scheme for a quarter-car with a PM-ASS that is controlled by the GP-FSB-ASMC+$H_\infty$, where the control law is shown in (33), the adaptive laws are Equations (34) and (35), and the sliding surface is Equation (23). The block "Road generator" simulates a reference road condition $y_m$ from a given terrain $R_r$ for the system. A Fuzzy logic controller (FLC) is used to track given terrain trajectories $R_r$. To compensate for the delay in the control input, a "Grey Model First-Order One Variable" state-predictor $GM(1,1)$ predicts the future error $\hat{e}$ for the FSB-ASMC+$H_\infty$, where $GM(1,1)$ is a time-series-forecasting model that predicts the possible error for the next sampling time. The steps for $GM(1,1)$ are:

Step 1: Define the original data set:

$$e^{(0)}(k) = \left\{e^{(0)}(1), e^{(0)}(2), \ldots, e^{(0)}(n)\right\}, \ n > 4 \tag{39}$$

Step 2: Do Accumulating Generation Operator (AGO) to generate an accumulated data set:

$$e^{(1)}(k) = \left\{e^{(1)}(1), e^{(1)}(2), \ldots, e^{(1)}(m)\right\} \tag{40}$$

where $e^{(1)}(k) = \sum_{i=1}^{k} e^{(0)}(i)$, $k = 1, 2, \ldots, m$.

Step 3: Construct a differential equation for $GM(1,1)$:

$$\frac{de^{(0)}}{dt} + ae^{(1)} = b \tag{41}$$

Step 4: Determine coefficients $a$ and $b$ using a Least Square Method (LSM):

$$\hat{a} = \begin{bmatrix} a \\ b \end{bmatrix} = (B^T B)^{-1} B^T Y \tag{42}$$

where $B = \begin{bmatrix} -\frac{1}{2}(e^{(0)}(1)+e^{(0)}(2)) & 1 \\ -\frac{1}{2}(e^{(0)}(2)+e^{(0)}(3)) & 1 \\ \vdots & \vdots \\ -\frac{1}{2}(e^{(0)}(n-1)+e^{(0)}(n)) & 1 \end{bmatrix}$ and $Y = \begin{bmatrix} e^{(0)}(2) & e^{(0)}(3) & \cdots & e^{(0)}(n) \end{bmatrix}^T$.

Step 5: Predict the next state as:

$$\hat{e}^{(0)}(k+1) = \hat{e}^{(1)}(k+1) - \hat{e}^{(1)}(k) \tag{43}$$

where $\hat{e}^{(1)}(k+1)$ is calculated using a whitening equation:

$$\hat{e}^{(1)}(k+1) = (e^{(1)}(1) - b/a) \times \exp(-ak) + b/a \tag{44}$$

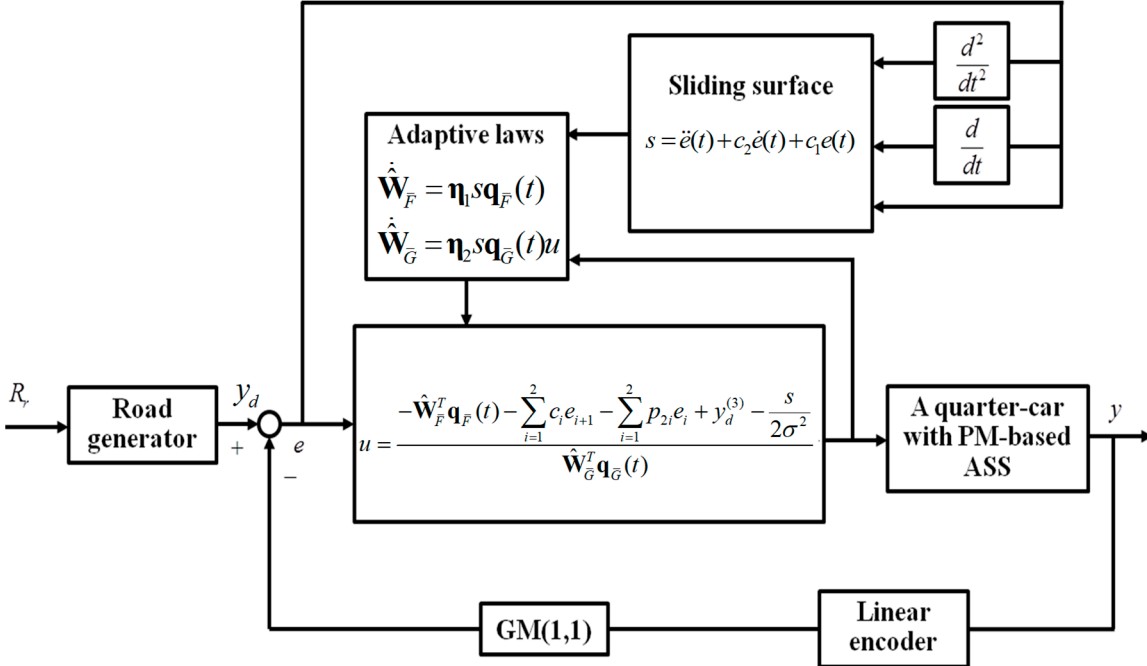

**Figure 4.** Diagram of the overall system for a quarter-car with a PM-ASS that is controlled by the GP-FSB-ASMC+$H_\infty$.

## 4. Experiments

Four different control strategies were applied on the test-rig: MacPherson struts, FSASMC, FSB-ASMC+$H_\infty$ and GP-FSB-ASMC+$H_\infty$. The ability of each strategy to reduce vibration was determined. When two Macpherson struts were used to suppress vibration, there was no additional force from the PM. FSB-ASMC means that the PM-ASS is operated on the FSB-ASMC; FSB-ASMC+$H_\infty$ means that the PMASS is operated on the FSB-ASMC with a $H_\infty$ performance guarantee and GP-FSB-ASMC+$H_\infty$ means that the PMASS is operated on the FSB-ASMC with a $H_\infty$ performance guarantee and $GM(1,1)$ error prediction. Two validations of the PM-ASS test-rig were performed:

(1) It was shown that the techniques for the $H_\infty$ performance guarantee and the state-predictor $GM(1,1)$ significantly improved the PM-ASS in terms of attenuating the displacement of the sprung mass and decreasing the acceleration of the sprung mass.
(2) The performance of the Macpherson struts and the FSB-ASMC with/without the $H_\infty$ performance guarantee and the state-predictor $GM(1,1)$ was compared.

### 4.1. Test-rig for the PM-ASS

Figure 5 shows the test-rig, which has a PM and two MacPherson struts installed in parallel between the vehicle body and the support frame, a wheel firmly installed on one side of the support frame and a road profile generator that generates road profiles for various road conditions. The road profile generator has rollers, an induction motor and a pneumatic cylinder, and is in close contact with the wheel. Figure 6 illustrates the system blocks for a quarter-car with a PM-ASS. The components in Figure 6 are detailed as follows:

(1) The sprung mass is the portion of the vehicle's total mass that is supported by the suspension, i.e., a chassis.
(2) The unsprung mass represents the brake, the caliper and ancillaries. The un-sprung mass does not include the mass of the wheel.
(3) Two MacPherson struts and one PM are used for the PM-ASS.
(4) A proportional pressure regulator (PPR) regulates the flow of air (pressure) into the PM.
(5) An inverter drives the induction motor to rotate the tire. The maximum speed of the tire is at 35 km/hr.
(6) A pneumatic cylinder generates various road conditions. The proportional directional control valve regulates the flow of air into the pneumatic cylinder. The pneumatic cylinder can lift a load of more than 400 kg and generates road profiles that are more than 10 cm high with a frequency of 10 Hz.
(7) On the plane of the support frame, one linear encoder and one accelerometer measures the displacement and the acceleration of the sprung mass.
(8) Two linear scalars measure the vertical variation in the unsprung mass and the road profile.
(9) The control $u$ is bounded within $[-5V, 5V]$.

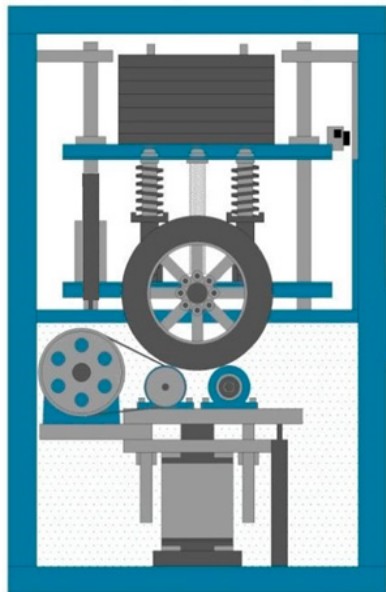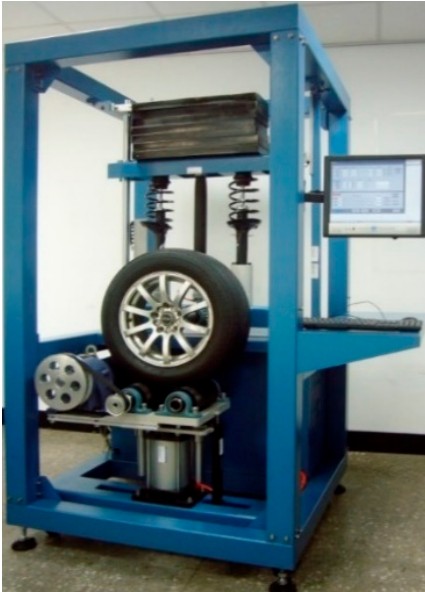

**Figure 5.** Test-rig of quarter-car using the PM-ASS.

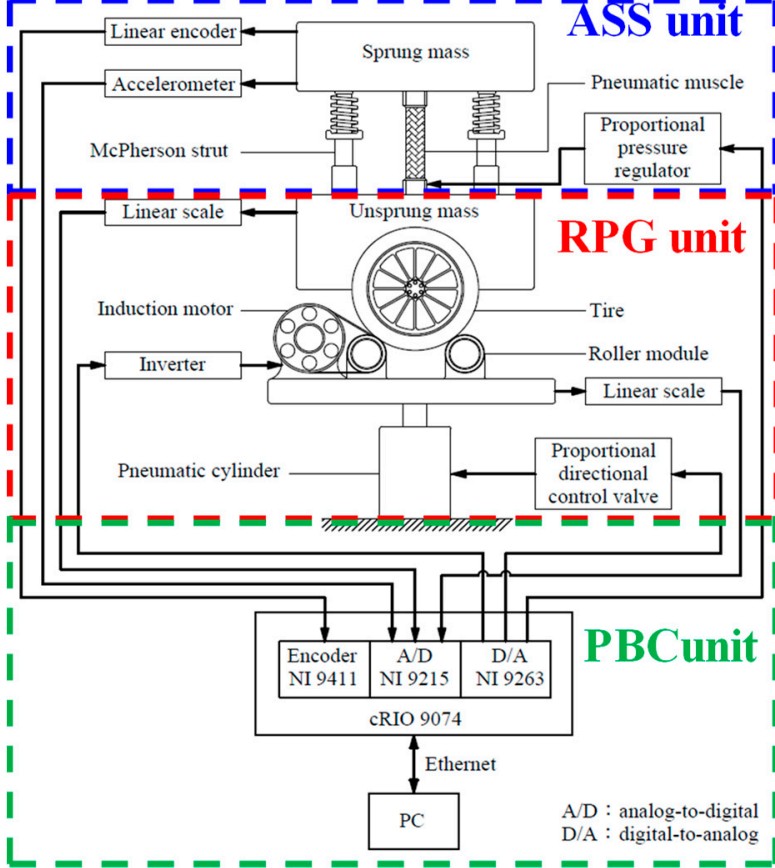

**Figure 6.** System blocks of quarter-car with the PM-ASS.

Figure 7 shows the mechatronics diagram for a quarter-car test-rig with a PM-ASS. The bottom of Figure 7 shows a control box, which has a NI cRIO-9074 embedded system, a NI-9263 D/A card, a NI-9215 A/D card and a NI-9411 encoder. The NI-9215 A/D card receives information (the green dashed line) about the displacement of the unsprung mass and the pneumatic cylinder and the acceleration of the sprung mass. The NI-9411 encoder receives a digital signal that shows the displacement of the sprung mass (the blue dashed line). The NI-9263AO sends the control signals (the red dashed line) to the PDVC, the invertor and the PPR to drive the PM, the pneumatic cylinder and the induction motor. All of the measured data was sent to a personal computer through an Ethernet interface and this is displayed on the screen. The system parameters can be modified in the PC and were sent to the NI cRIO-9074 embedded system. The Butter Worth filter was adopted to attenuate the sensor noise for the linear encoders, which is expressed as:

$$y_{out}(k) = 0.97y_{out}(k-1) + 0.015[y_{in}(k) + y_{out}(k-1)] \tag{45}$$

where $y_{out}(k)$ is the output of the filter at the *k*th sampling and $y_{in}(k)$ is the displacement of the sprung mass measured by the linear encoder at the *k*th sampling.

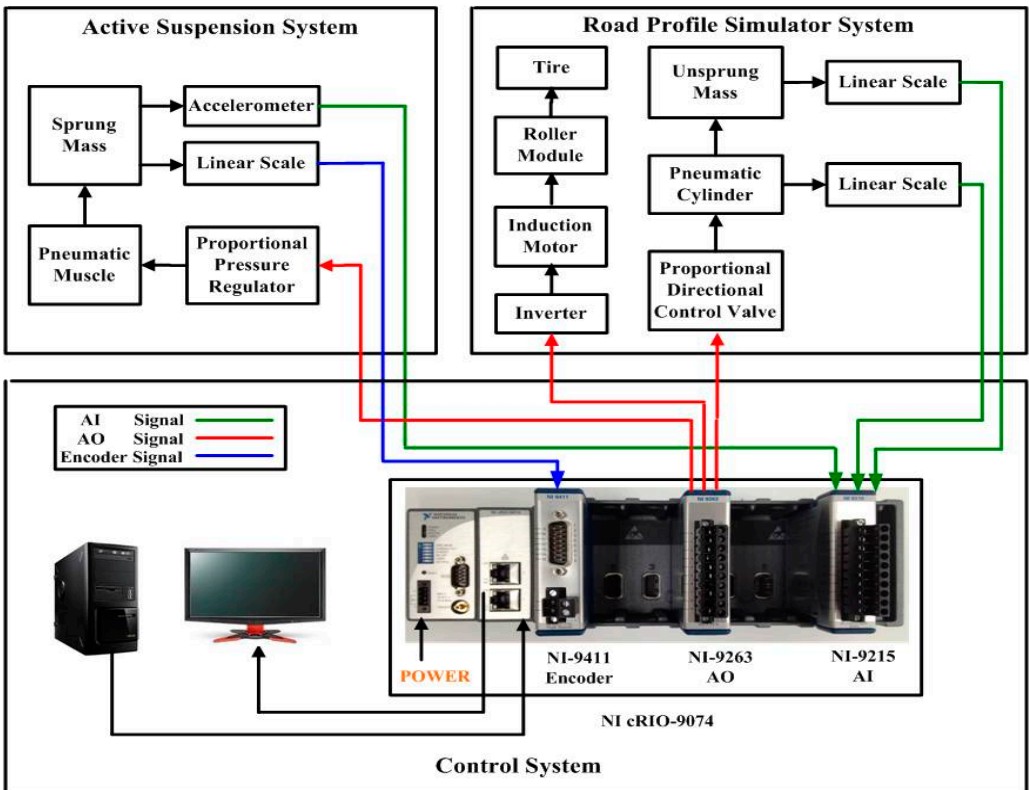

**Figure 7.** Mechatronics diagram of the quarter-car test-rig with the PM-ASS.

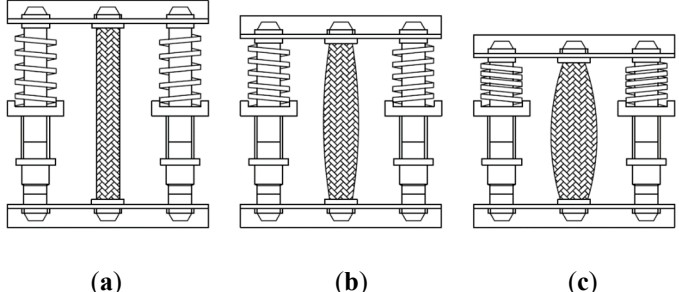

|(**a**)|(**b**)|(**c**)|

**Figure 8.** States of the PM (**a**) no-expansion, (**b**) half-expanded, (**c**) fully-expanded.

*4.2. Initial Setups for Experiments*

It is critically important that the PM-ASS reacts quickly to changes in road conditions, thus, the PM was initially pre-set in the half-expanded state. The initial voltage of the PPR was 5V, in the range of [0V 10V] to maintain the PM in the half-expanded condition. Figure 8 shows three states for the PM: no-expansion (Figure 8a), the half-expanded state (Figure 8b) and the fully expanded state (Figure 8c). A rough concave-convex road profile and a two-bump excitation road profile were used to verify stability and robustness of the proposed PM-ASS. A fuzzy logic control (FLC) tracked the road profiles. Five membership functions were used for the error, the change in the error and the fuzzy output, as shown in Figure 9. The fuzzy rule refers to the rule template in Chapter of [25], and the center of maximum defuzzification was used to calculate the fuzzy output. Table 1 shows the parameters for the FSB-ASMC, the FSB-ASMC+$H_\infty$ and the GP-FSB-ASMC+$H_\infty$.

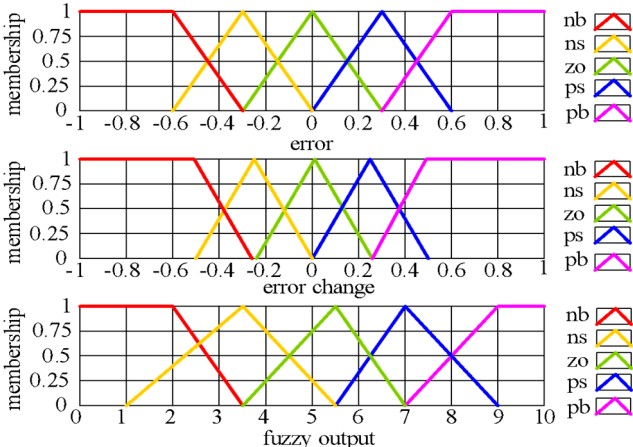

**Figure 9.** Fuzzy partitions and membership functions for the error, the change in error and the fuzzy output.

**Table 1.** Parameters for the proposed method for a rough concave-convex road profile.

| Initial Weights | Control Parameters |
|---|---|
| $W_F = \begin{bmatrix} 0.053, 0.003, 0.021, 0.01, 0.038, 0.02, 0.01, -0.007 \\ -0.042, 0.017, 0.04, -0.01, 0.014, 0.025, 0.01, 0.05 \end{bmatrix}$ | $c_1 = 3.2$<br>$c_2 = 3.3$<br>$A_{21} = 0.1563$<br>$A_{22} = 0.8049$ |
| $W_G = \begin{bmatrix} 5.08, 0.01, 0.017, 0.01, 0.007, 0.013, 0.02, 0.011 \\ 0.018, 0.011, 0.007, 0.021, -0.003, 0.008, 0.011, 0.013 \end{bmatrix}$ | $\sigma = 0.5$<br>$\eta_1 = 1$<br>$\eta_2 = 0.01$ |

*4.3. Results and Disscussion of Experiments*

**Experiment 1.** *Vehicle Riding on a Rough Concave-Convex Road Profile.*

The rough concave-convex road profile is composed of a bump and a hollow with a sinusoidal disturbance, which is represented Equation (46).

$$R_r(t) = \begin{cases} -1.554(t-3.5)^3 + 3.5(t-3.5)^2 + d(t) & for\, t \in [3.5, 5) \\ 1.554(t-6.5)^3 + 3.5(t-6.5)^2 + d(t) & for\, t \in [5, 6.5) \\ 1.554(t-8.5)^3 - 3.5(t-8.5)^2 + d(t) & for\, t \in [8.5, 10) \\ -1.554(t-11.5)^3 - 3.5(t-11.5)^2 + d(t) & for\, t \in [10, 11.5) \\ d(t) & else \end{cases} \quad (46)$$

where the sinusoidal disturbance is $d(t) = 0.175 \sin(2\pi t) + 0.07 \sin(7.5\pi t)$. Figure 10 shows the variation in the displacement of the sprung mass when the quarter car travels on a rough concave-convex road with immediate 3 cm changes. In Figure 10, the blue line shows that the road profiles that are generated by the Road Generator have a ±0.5 cm error in the rough concave-convex road profile. The pink dotted line shows the displacement of the sprung mass, which is controlled by the MacPherson struts. Because no additional force is supplied from PM in this case, the maximum displacement of the sprung mass is 2.4 cm, thus, only a 0.6 cm-high vibration is suppressed. Figure 10 shows that the FSB-ASMC, the FSB-ASMC+$H_\infty$ and the GP-FSB-ASMC+$H_\infty$ better suppress vibration than MacPherson struts under the same conditions. The black line in Figure 10 shows that FSB-ASMC suppresses a vibration in the sprung mass of less than 1.2 cm. The green dashed line shows that the FSB-ASMC+$H_\infty$ suppresses a maximum vibration of less than 0.65 cm because it attenuates lumped uncertainties. The GP-FSB-ASMC+$H_\infty$ limits the maximum vibration to less than 0.4 cm, which is

better than either the FSB-ASMC or the FSB-ASMC+$H_\infty$. Table 2 compares the maximum vibration and the root-mean-square error (RMSE) for the vibration, where the RMSE is defined as:

$$RMSE = \sqrt{\frac{1}{N}\sum_{i=1}^{N}\left(value_{vibration} - value_{zero\_level}\right)^2} \qquad (47)$$

where $value_{vibration}$ is the measured vibration, $value_{zero\_level}$ is the zero displacement of the sprung mass and $N$ is the total number of the measured vibration data points. Table 3 shows the weights after training for the GP-FSB-ASMC+$H_\infty$. In Figure 10, there is a slight oscillation on the sprung mass because of tire deformation, which is a small variation that depends on the road profile. Using the FLC, the variation in the output of the road generator is shown as a blue line in Figure 10, for which the tracking error is limited to ±0.5 cm. The acceleration and the control voltage and the sprung mass for all methods are respectively shown in Figures 11 and 12. It is noted that no additional force is applied to MacPherson struts, thus, Figure 12 shows no control voltage for them.

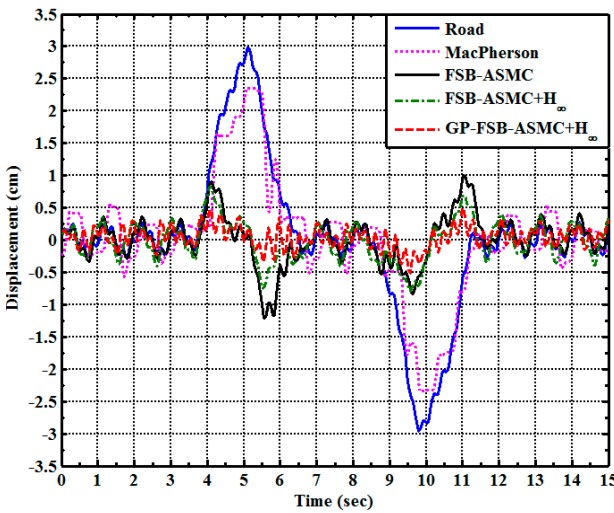

**Figure 10.** Displacement of the sprung mass for a rough concave-convex road profile.

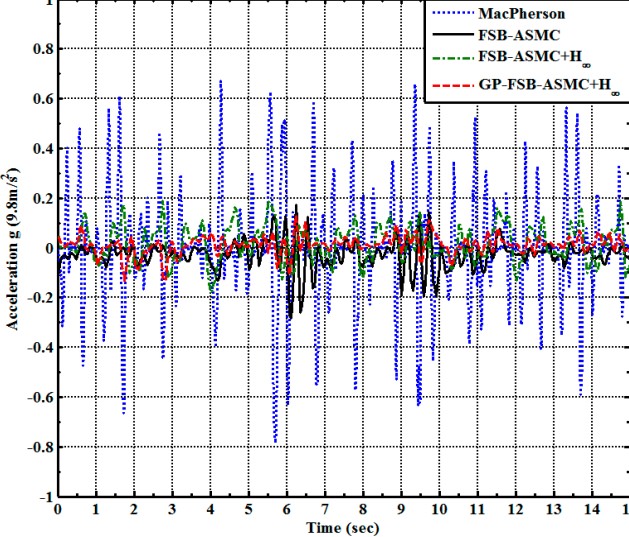

**Figure 11.** Acceleration of the sprung mass for the rough concave-convex road profile.

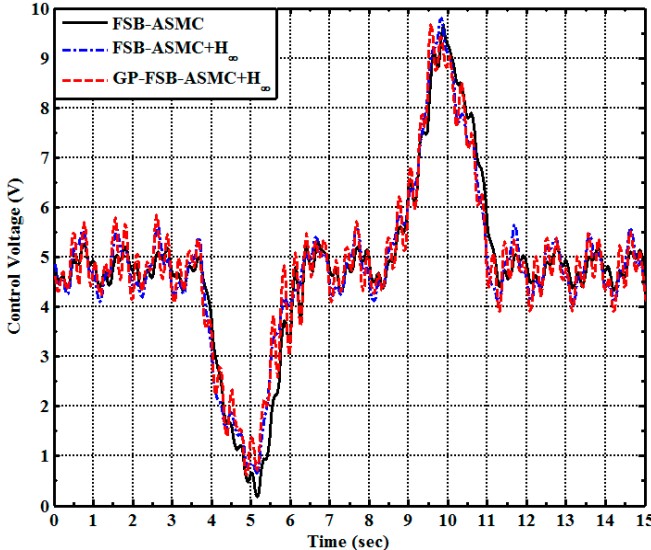

**Figure 12.** Control voltage for the rough concave-convex road profile.

**Table 2.** Performance comparisons of the PM-ASS with four different control strategies for the rough concave-convex road profile.

| Control Strategies | Maximum of Displacement (cm) | Acceleration (cm/s$^2$) |
|---|---|---|
| MacPherson | 1.909 | 1.165 |
| FSB-ASMC | 1.006 | 0.397 |
| FSB-ASMC+$H_\infty$ | 0.838 | 0.224 |
| GP-FSB-ASMC+$H_\infty$ | 0.593 | 0.112 |

*Performances / Control Strategies*

**Table 3.** Trained weights for the GP-FSB-ASMC+$H^\infty$ for the rough concave-convex road profile.

**Trained Weights**

$$W_{\overline{F}} = \begin{bmatrix} 0.014, 0.458, -0.014, 0.183, 0.014, 0.413, -0.14, 0.22 \\ -0.14, 0.15, 0.016, -0.01, 0.07, 0.206, -0.01, 0.293 \end{bmatrix}$$

$$W_{\overline{G}} = \begin{bmatrix} 1.8, 0.029, -0.07, 0.0917, 0.0206, 0.007, -0.367, -0.059 \\ -0.079, 0.573, -0.825, 0.007, 0.15, 0.012, -0.07, 0.147 \end{bmatrix}$$

**Experiment 2.** *Vehicle Riding on a Sine Wave Road Profile.*

A sine wave excitation road profile is represented as:

$$R_r(t) = 1.5\sin(\pi t) \quad for \quad t \in [0, 10] \tag{48}$$

Figure 13 shows the variation in the displacement of the sprung mass when the quarter car moves along a rough sine wave excitation with a 1.5 cm amplitude and 2.5-sec wavelength. In Figure 13, the blue line shows the road profiles that are generated by the Road Generator, which gives a ±0.5 cm error in the rough sine wave excitation road profile. The pink dotted line shows the displacement of the sprung mass, which is controlled by the MacPherson struts. There is almost no vibration-reduction. The black line in Figure 13 shows that FSB-ASMC suppresses the vibration of the sprung mass to less than 0.6cm. The green dashed line shows that the FSB-ASMC+$H_\infty$ suppresses the maximum vibration to less than 0.6 cm. The GP-FSB-ASMC+$H_\infty$ limits the maximum vibration to less than 0.25 cm, which is better than either the FSB-ASMC or the FSB-ASMC+$H_\infty$. Table 4 compares the maximum vibration and the root-mean-square error (RMSE) of the vibration. Table 5 shows the weights after training for

the GP-FSB-ASMC+$H_\infty$. Using the FLC, the variation in the output of the road generator is shown as the blue line in Figure 13 for which the tracking error is limited to ±0.1 cm. The acceleration and the control voltage and the sprung mass for all methods are, respectively, shown in Figures 14 and 15.

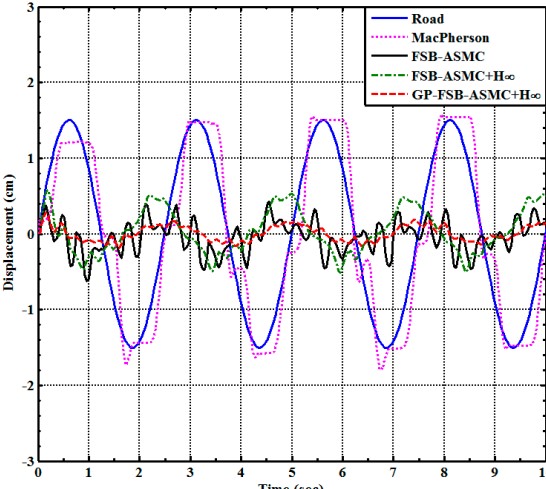

**Figure 13.** Displacement of the sprung mass for the sine wave excitation road profile.

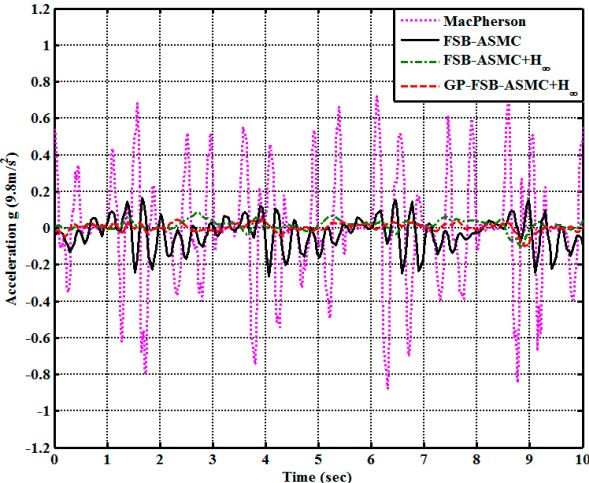

**Figure 14.** Acceleration of the sprung mass for the sine wave excitation road profile.

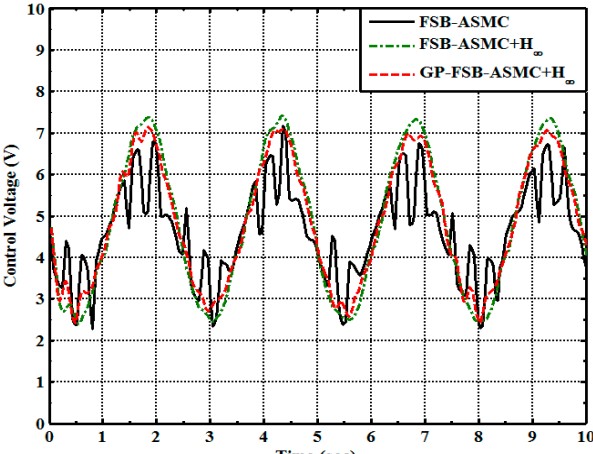

**Figure 15.** Control voltage for the sine wave excitation road profile.

**Table 4.** Performance comparisons of the PM-ASS with four different control strategies for the rough concave-convex road profile.

| Performances Control Strategies | Maximum of Displacement (cm) | Acceleration (cm/s$^2$) |
|---|---|---|
| MacPherson | 1.571 | 1.604 |
| FSB-ASMC | 1.06 | 0.546 |
| FSB-ASMC+$H_\infty$ | 0.581 | 0.187 |
| GP-FSB-ASMC+$H_\infty$ | 0.285 | 0.119 |

**Table 5.** Trained weights for the GP-FSB-ASMC+$H_\infty$ for the rough concave-convex road profile.

| Trained Weights |
|---|
| $W_{\overline{F}} = \begin{bmatrix} 0.014, 0.458, -0.014, 0.183, 0.014, 0.413, -0.14, 0.22 \\ -0.14, 0.15, 0.016, -0.01, 0.07, 0.206, -0.01, 0.293 \end{bmatrix}$ |
| $W_{\overline{G}} = \begin{bmatrix} 1.45, 0.019, -0.085, 0.1, 0.05, 0.008, -0.257, -0.051 \\ -0.059, 0.516, -0.596, 0.056, 0.23, 0.1, -0.02, 0.147 \end{bmatrix}$ |

## 5. Conclusions

This study developed an experimental ASS, which was termed the PM-ASS, on a quarter-car test rig. The design of the PM-ASS used a PM that is installed in parallel with two MacPherson struts and can provide a stable vertical force to reduce vibrations. In terms of the controller design, a GP-FSB-ASMC+$H_\infty$ was used for the PM-ASS to address the nonlinearity and time-varying characteristics. Benefiting from the use of the orthogonal Fourier basis function, the FSB-ASMC+$H_\infty$ has a clear physical meaning and readily determined structure for approximating the optimal control law to attenuate disturbances and uncertainties. In addition, the state-predictor $GM(1,1)$ predicted future errors to enhance the control performance for the FSB-ASMC+$H_\infty$. The experimental results demonstrated that the PM-ASS with the GP-FSB-ASMC+$H_\infty$ significantly improved the Macpherson Strut suspension in terms of suppression of the displacement and acceleration in the vehicle body for improving the ride comfort.

To enhance the ability of vibration-reduction for the PM-ASS, we advise future researchers to use two air intakes on both side of the PM. A faster control response will then be produced for the PM-ASS. In terms of the GP-FSB-ASMC+$H_\infty$, a systemic method can be further developed for determining the initial weights and the controller parameters.

## 6. Patents

There are three Taiwan utility model patents resulting from the developed active suspension system, which are (1) a double-layer vibration isolation system using the pneumatic muscles with the patent number M437402; (2) a support system for the pneumatic muscles with the patent number M437403; (3) active vehicle suspension system using the pneumatic muscles with the patent number M438403.

**Author Contributions:** Conceptualization, I.-H.L. and L.-W.L.; methodology, L.-W.L. and I.-H.L.; software, I.-H.L.; validation, L.-W.L. and I.-H.L.; formal analysis, L.-W.L.; investigation, I.-H.L. and L.-W.L.; resources, L.-W.L.; data curation, L.-W.L.; writing—original draft preparation, I.-H.L.; writing—review and editing, I.-H.L. and L.-W.L.; supervision, I.-H.L. and L.-W.L.; project administration, L.-W.L.; funding acquisition, L.-W.L.

**Funding:** This research has received financial support from the Ministry of Science and Technology, R. O. C. Grant: No. MOST 107-2221-E-005-077-, 106-2221-E-032-060-MY2, 108-2634-F-003-003, 108-2634-F-003-002, 108-2221-E-034-017, 108-2634-F-003-003, 108-2628-E-005-003-MY2, and NSC 100-2221-E-262-012.

**Acknowledgments:** The authors would like to thank reviewers and the editor for their helpful and detailed comments that have assisted in improving the presentation of this paper.

**Conflicts of Interest:** The authors declare no conflict of interest.

## Appendix A

**Proof of Lemma 1:** Consider the Lyapunov Function:

$$V = \frac{1}{2}s^2 + \frac{1}{2}\mathbf{e}^T\mathbf{P}\mathbf{e} \tag{A1}$$

Differentiating Equation (A1) with respective to $t$ yields:

$$\dot{V} = s\dot{s} + \frac{1}{2}\dot{\mathbf{e}}^T\mathbf{P}\mathbf{e} + \frac{1}{2}\mathbf{e}^T\mathbf{P}\dot{\mathbf{e}} \tag{A2}$$

Substituting Equations (25) and (34) into Equation (A2) yields:

$$
\begin{aligned}
\dot{V} &= s\dot{s} + \frac{1}{2}\{\mathbf{e}^T\mathbf{A}_1^T + [0,s]\}\mathbf{P}\mathbf{e} + \frac{1}{2}\mathbf{e}^T\mathbf{P}\{\mathbf{A}_1\mathbf{e} + [0,s]^T\} \\
&= s\dot{s} + \frac{1}{2}\mathbf{e}^T[\mathbf{A}_1^T\mathbf{P} + \mathbf{P}\mathbf{A}_1]\mathbf{e} + [0,s]\mathbf{P}\mathbf{e} \\
&= s[e^{(3)} + c_2\ddot{e} + c_1\dot{e}] - \frac{1}{2}\mathbf{e}^T\mathbf{Q}_1\mathbf{e} + [0,s]\mathbf{P}\mathbf{e} \\
&= s[y^{(3)} - y_d^{(3)} + c_2\ddot{e} + c_1\dot{e}] - \frac{1}{2}\mathbf{e}^T\mathbf{Q}_1\mathbf{e} + [0,s]\mathbf{P}\mathbf{e} \\
&= s[\overline{F}(\mathbf{z}) + \overline{G}(\mathbf{z})u + d(t) - y_d^{(3)} + c_2\ddot{e} + c_1\dot{e}] - \frac{1}{2}\mathbf{e}^T\mathbf{Q}_1\mathbf{e} + s\sum_{i=1}^{2} p_{2i}e_i
\end{aligned}
\tag{A3}
$$

Apply Equation (26) to (A3) and let $k = k_1 + d^u$, $k_1 > 0$. We have the following relationship:

$$\dot{V} = -k|s| - \frac{1}{2}\mathbf{e}^T\mathbf{Q}_1\mathbf{e} + sd(t) \le -k_1|s| - \frac{1}{2}\mathbf{e}^T\mathbf{Q}_1\mathbf{e} \le 0. \tag{A4}$$

According to Barbarlet's Lemma, $s \to 0$ and $\mathbf{e} \to 0$ as $t \to \infty$. This completes the proof. □

## Appendix B

**Proof of Theorem 1:** For the Lyapunov function:

$$\mathrm{V} = \frac{1}{2}s^2 + \frac{1}{2}\mathbf{e}^T\mathbf{P}\mathbf{e} + \frac{1}{2}\widetilde{\mathbf{W}}_{\overline{F}}^T\boldsymbol{\eta}_1^{-1}\widetilde{\mathbf{W}}_{\overline{F}} + \frac{1}{2}\widetilde{\mathbf{W}}_{\overline{G}}^T\boldsymbol{\eta}_2^{-1}\widetilde{\mathbf{W}}_{\overline{G}} \tag{A5}$$

Differentiating Equation (A5) yields:

$$\dot{V} = s\dot{s} + \frac{1}{2}\dot{\mathbf{e}}^T\mathbf{P}\mathbf{e} + \frac{1}{2}\mathbf{e}^T\mathbf{P}\dot{\mathbf{e}} + \widetilde{\mathbf{W}}_{\overline{F}}^T\boldsymbol{\eta}_1^{-1}\dot{\widetilde{\mathbf{W}}}_{\overline{F}} + \widetilde{\mathbf{W}}_{\overline{G}}^T\boldsymbol{\eta}_2^{-1}\dot{\widetilde{\mathbf{W}}}_{\overline{G}} \tag{A6}$$

Substituting Equation (32) and Equation (34) into Equation (A6) yields

$$
\begin{aligned}
\dot{V} &= s[\hat{\mathbf{W}}_{\overline{F}}^T\mathbf{q}_{\overline{F}}(t) + \hat{\mathbf{W}}_{\overline{G}}^T\mathbf{q}_{\overline{G}}(t)u + w_t - y_d^{(3)} + c_2\ddot{e} + c_1\dot{e}] - \frac{1}{2}\mathbf{e}^T\mathbf{Q}\mathbf{e} + s\sum_{i=1}^{2} p_{2i}e_i \\
&\quad + \widetilde{\mathbf{W}}_{\overline{F}}^T(s\mathbf{q}_{\overline{F}}(t) - \boldsymbol{\eta}_1^{-1}\dot{\hat{\mathbf{W}}}_{\overline{F}}) + \widetilde{\mathbf{W}}_{\overline{G}}^T(s\mathbf{q}_{\overline{G}}(t)u - \boldsymbol{\eta}_2^{-1}\dot{\hat{\mathbf{W}}}_{\overline{G}})
\end{aligned}
\tag{A7}
$$

Substituting Equation (33) and Equations (35)–(36) into Equation (A7) yields:

$$\dot{V} = -\frac{1}{2}\mathbf{e}^T\mathbf{Q}_1\mathbf{e} - \frac{1}{2}\left(\frac{s}{\sigma} - \sigma w_t\right)^2 + \frac{1}{2}\sigma^2 w_t^2 \le -\frac{1}{2}\mathbf{e}^T\mathbf{Q}_1\mathbf{e} + \frac{1}{2}\sigma^2 w_t^2 \tag{A8}$$

Integrating Equation (A8) from $t = 0$ to $t = T$ yields:

$$\int_0^T \dot{V}(\tau)d\tau \le -\frac{1}{2}\int_0^T \mathbf{e}^T(\tau)\mathbf{Q}_1\mathbf{e}(\tau)d\tau + \frac{1}{2}\sigma^2\int_0^T w_t^2 d\tau \tag{A9}$$

and:

$$\frac{1}{2}\int_0^T \mathbf{e}^T(\tau)\mathbf{Q_1}\mathbf{e}(\tau)d\tau \leq \mathrm{V}(0) + \frac{1}{2}\sigma^2\int_0^T w_t{}^2 d\tau \tag{A10}$$

Substituting Equation (A5) into Equation (A10) gives the $H_\infty$ tracking performance, which is:

$$\frac{1}{2}\int_0^T \mathbf{e}^T(\tau)\mathbf{Q_1}\mathbf{e}(\tau)d\tau \leq \frac{1}{2}s^2(0) + \frac{1}{2}\mathbf{e}^T(0)\mathbf{A}\mathbf{e}(0) + \frac{1}{2}\widetilde{\mathbf{W}}_{\overline{F}}^T(0)\mathbf{\eta}_1^{-1}\widetilde{\mathbf{W}}_{\overline{F}}(0) + \frac{1}{2}\widetilde{\mathbf{W}}_{\overline{G}}^T(0)\mathbf{\eta}_2^{-1}\widetilde{\mathbf{W}}_{\overline{G}}(0) + \frac{1}{2}\sigma^2\int_0^T w_t{}^2 d\tau \tag{A11}$$

This completes the proof. □

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
