# Peer review of "Design and Development of an Active Suspension System Using Pneumatic-Muscle Actuator and Intelligent Control"

_applsci, doi:10.3390/app9204453_

Round 1
Reviewer 1 Report
Very interesting and well written paper regarding the pneumatic-muscle actuator and intelligent control which can be applied into the improvement of the rid comfort. A systematic mathematical model calculation and experiments were detail demonstrated in this paper.
Detail Comments as below:
Page 1, Lines 21: Please give a definition of GM (1, 1) Lines 36-39: If those statement came from any paper, please include the reference information. Page3, I would like to shift line 110-121 to the end section. Not really related with the section of the mathematical model.
Based on the quarter-car experiment test results did prove the advantages of the proposed controller help on reduce (not cancel) the car’s sprung vibrations. This impact can predict this will bring benefit of the comfort of driving cars but yet to implement it to real car testing. Overall a good paper and need some minor revise and improvement.
Author Response
The paper we submitted to Applied Sciences entitled, “Design and Development of an Active Suspension System Using Pneumatic-Muscle Actuator and Intelligent Control” that I wish to be considered for publication in the Applied Sciences. All coauthors of this paper contributed to the work and approved contents of the final version before submission. This manuscript has not been accepted for publication elsewhere. I would appreciate it if you could expedite the review process for this manuscript.
Thank you very much for your kindly reminder. The replies are enclosed in the attachment.

Reviewer 2 Report
The work is well structured and described. However, I do expect more details to discuss the results. In addition, conclusions must be reformulated such as to introduce the key findings, limitations, and the future work intended to improve the current results.
Author Response

(The authors gave the same response as above.)

Reviewer 3 Report
The paper under review considers the issue of a design and development
of an active suspension system using pneumatic-muscle actuator and intelligent control
In the reviewer’s opinion, in general, the paper is quite interesting.
However, there are several important aspects that require authors comments and/or improvement:
1) Line 13: "regulate" should be control
2) Line 21: "GM(1,1)" it is necessary to mention this?
3) Line 46: al.[7] should be control al. [7]
4) Line 81 to 95: How text between this line is related to the next 5 pints?
5) Line 110 to 121: This test if from template...
6) Eq. 16: The F and G are provided but later are unknown, please explain.
7) Line 198: Put equation from this line in the new equation as for example 19. In math mode not in inline mode.
8) Section 3. Where is the Hinf tracking performance in this section?
9) Line 238: In Lyapunov equation typically the matrix P is positive definite and unknown not matrix A.
Matrix A is a typical system matrix in state-space form system. I suggest you change it in all equations.
10) Line 238, 241: Use \succ instead of > symbol.
11) Line 250: Move text to new page
12) Line 261: put cos w_0 X as cos(w_0X) for all elements in a vector.
13) Line 284: u \in [-5V 5V] should be control u \in [-5V, 5V] and move this to Experiments sextion
14) Line 285: state are bunded. What is value of region that state are bunded.
15) Line 357: Provide the final design procedure to help reader before section 4. Divide into offline and online mode.
16) Fig 4. The diagram can be move before section 4 to help understand the control strategy.
17) Line 429: Move text to new newpage.
Author Response

(The authors gave the same response as above.)

Round 2
Reviewer 3 Report
The paper under review considers the issue of a design and development
of an active suspension system using pneumatic-muscle actuator and intelligent control. The answers to my comments provided by the authors are satisfactory.
Finally, I have one point: you should format the text thoroughly and well!!!. Eg on page 12 the diagram is not fully visible !!!
Author Response
Dear Reviewer:
The paper we submitted to Applied Sciences entitled, “Design and Development of an Active Suspension System Using Pneumatic-Muscle Actuator and Intelligent Control” that I wish to be considered for publication in the Applied Sciences. All coauthors of this paper contributed to the work and approved contents of the final version before submission. This manuscript has not been accepted for publication elsewhere. I would appreciate it if you could expedite the review process for this manuscript.
Thank you very much for your kindly reminder. The replies are attached.
